

# Long-term Effects of Drainage and Rewetting on the Degradation and Preservation of Peat Organic Matter in Warm Climate

Guy Sapir[1,2,3], Alon Angert[1], Yoav Oved Rosenberg[2], Rotem Golan[3].

[1] The Institute of Earth Sciences, The Hebrew University of Jerusalem, Israel. [2] Geological Survey of Israel, Jerusalem, Israel.

[3] Institute of Soil, Water and Environmental Sciences. Volcani Institute, Agricultural Research Organization, Israel.

*Correspondence to*: Alon Angert (angert@huji.ac.il)

**Abstract.** Peatlands cover about 3% of the earth's land surface, while storing about 20% of the total global soil organic carbon. These carbon stocks are largely at risk as many peatlands have deteriorated since the Industrial Revolution due to conversion to agricultural land by drainage. Globally, peatland drainage is responsible for over 3.5% of anthropogenic greenhouse gas emissions. About 75% of these emissions originate from warm climate regions. Mitigation of these emissions can be achieved by rewetting degraded peatlands. This study focuses on a warm-climate peatland that has been cultivated for the past ~70 years (Hula Valley, Israel). The historic marsh was drained in 1957 for agricultural use and underwent a hydrological restoration project for elevating and stabilizing groundwater table since 1994. This land management history resulted in a sedimentary peat column that can be divided into three distinct sub-sections: drained, rewetted and pristine peat. This setting enables studying the drainage and rewetting effects on soil organic matter (SOM) degradation and preservation under warm climates. For this purpose, five sediment cores, 4 m long each, were excavated from cropland located over the historic marsh area. Locations were chosen to match previous studies on this site. Each soil profile was characterized using Rock-Eval® thermal analysis of the organic matter, and short-term soil aerobic respiration experiments. Integration of these results with historic SOM content data and with SOM modelling was used to explore the long-term process and rate of degradation. We found that the mean SOM content in the top one meter of the soil profile declined from 68 ±4 wt.% to 21 ±2 wt.% over the past 66 years, excluding compaction effect. In comparison to the drained section, the rewetted and pristine sub-sections has a mean SOM of 33 ±2 wt.% and 64 ±2 wt.%, respectively. A peak in pyrite concentration beneath the recent water table-level, was observed in most profiles, indicating anaerobic conditions and sulfur recycling. Rock-Eval® thermal analysis demonstrated that during decomposition, the residual SOM became more oxidized and contained a lower proportion of thermally labile SOM, with a significant difference found between drained and rewetted peat. These results imply that the raising of the water table (~30 years ago) effectively helped preserving organic matter compared to the





drained section. Long-term SOM field data were integrated and studied using an SOM decomposition model and by incorporating respiration fluxes. The resulting trends highlighted that the first few decades of exposure are highly significant for the fate of the carbon stock, leading to substantial $CO_2$ emissions. These emissions were lower by 60–85% after 70 years. Furthermore, our results suggest that currently, approximately 13-21 wt.% of SOM persists as resistant organic matter in the degraded peat.

## 1    Introduction


Peatlands cover about 3% of the earth's land surface and estimated to store between 500-700 Pg of carbon (C), which are about 20% of the global total soil organic stock (Page et al., 2016). Peatland has been exploited intensively by humans since the Industrial Revolution. At present, human activity of draining and mining affects approximately 10% of the remaining global peatlands, transforming these ecosystems from long-term carbon

sinks into significant sources of greenhouse gases (GHG's) (Leifeld & Menichetti, 2018). This transformation promotes carbon loss, primarily through microbial respiration and dissolved organic carbon (DOC) leaching (Hiraishi et al., 2013). Degraded peatlands currently contain 80.8 Pg of carbon and 2.3 Pg of nitrogen (Leifeld & Menichetti, 2018). Without further protective actions, drained peatlands will release most of this stock as greenhouse gases into the atmosphere.

The primary factor that contributes to the preservation of peat organic matter (OM) is anaerobic conditions, which predominantly prevail in water-saturated conditions (Moore et al., 1989). In such environments, where the SOM supply rate exceeds the decomposition rate, a net carbon sink can evolve (Moore, 1989). Generally, organic matter is added to the surface of the mire (peat forming habitat) mainly from net primary production, and its accumulation is countered by decomposition processes. Decomposition in the top layer occurs under aerobic

conditions, followed by deeper burial in the saturated horizon where oxygen is limited (Moore 1989; Joosten, 2002). There, under anaerobic conditions, decomposition occurs at rates that are approximately a thousand times lower than the upper aerobic layer (Clymo et al., 1998; Kleinen et al., 2012). This low decomposition rates result in a carbon sink with a turnover time of several thousands of years if remains undisturbed (Yu et al., 2010). Lowering the water table level (WTL) results in oxidation of the peat column, stimulating aerobic microbial

decomposition of the SOM (Scanlon et al., 2000; Joosten, 2002), which releases intense $CO_2$ and $N_2O$ fluxes into the atmosphere (Page et al., 2016). Draining peatlands by lowering the WTL to convert them into agricultural lands has been a widespread practice for centuries and continues today. Numerous studies, e.g., Hooijer et al. (2010) and Moore et al. (1989), have demonstrated a clear correlation between low WTL and high $CO_2$ emissions



due to peat soil respiration. Currently, more than 90% of global drained peatlands are used for agriculture
(Tubiello et al., 2016). This practice occurs across most climate zones, with the largest emissions rates from
drained peatland are currently in Southeast Asia (Hooijer et al., 2010; Tubiello et al., 2016). Emissions rates due
to microbial respiration are strongly controlled by temperature and therefore by climate zones. Drained peatlands
in tropical regions have the highest annual emissions per area (~ 70-ton $CO_2$ ha$^{-1}$ yr$^{-1}$) which is four and two times
higher than in boreal and warm temperate climate zones, respectively (Tubiello et al., 2016, Hiraishi et al., 2013).
Global emissions stemming from drained peatlands are estimated to be ~1.9 Pg $CO_2$-eq yr$^{-1}$ (Leifeld &
Menichetti, 2018), accounting for ~3.6% of global anthropogenic $CO_2$-eq emissions, with ~77% of those
emissions coming from tropical climate zone (Leifeld & Menichetti, 2018).

Peatland restoration, primarily through rewetting drained peat, can reduce global $CO_2$ emissions by 0.5-1.3 Pg
$CO_2$-eq yr$^{-1}$ (Leifeld and Menichetti, 2018; Gunther et al., 2020), and account for 30% of global mitigation
solutions for forests and other land ecosystems (Roe et al., 2021). Successful rewetting often involves blocking
drainage systems, stabilizing water levels, and re-establishing vegetation adapted to wet conditions (Convention
on Wetlands. 2021). While drainage leads to high $CO_2$ emissions, maintaining a high WTL can result in increased
methane emissions. However, studies showed that raising the WTL to near the surface results in a net reduction
of $CO_2$ emissions that outweighs methane emissions (Gunther et al., 2020; Wilson et al., 2016; Nyberg et al.,
2022). A global meta-analysis found that keeping the groundwater level close to the surface can reduce global
GHG's emissions from wetlands by about 90% by the end of the 21st century (Zou et al., 2022). Despite many
studies highlighting the importance of rewetting for climate mitigation, there is scarce data on organic matter
preservation in rewetted tropical peatlands (Nabuurs et al., 2023; Wilson et al., 2016). Available data on rewetted
sites are limited for several to ten years (Wilson et al., 2016), thus limiting the evaluation of rewetting effect on
SOC preservation and GHG's emissions. Moreover, information about the time-frame since drainage, during
which rewetting is effective, is crucial for selecting sites for management and preservation actions.

Peatlands deposits are rich in organic matter, characterized by about 50 wt.% SOM (Joosten, 2002). While
drainage leads to SOM loss and mineralization, studies have shown that recalcitrant material in peatlands may
influence decomposition rates after drainage (Leifeld et al., 2012). Leifeld et al. (2012) showed that, in drained
peatlands, high respiration rates were correlated with high concentrations of O-alkyl-C (a functional group of
labile compounds). Könönen et al. (2016) found that tropical peat drainage leads to an increase in recalcitrant
compounds concentrations such as lignin. Phenol oxidase is known as a key factor in accelerating the
decomposition of recalcitrant SOM such as lignin after drainage (Freeman et al., 2004). However, Fe(II)
oxidation may inhibit phenol oxidase activity and further enhancing Fe(III) and lignin complexation, as suggested



by the 'iron gate' mechanism (Wang et al., 2017). These studies attest that the reactivity for decomposition depends on the different fractions and compounds of the SOM in peat. However, the fate of these recalcitrant compounds after prolonged peat degradation and mineralization is yet unknown.

Recent evidence in mineral soils studies suggests that stabilization by recalcitrant material, as proposed in the 'selective preservation' hypothesis, is not common and relevant only at the beginning of SOM decomposition and
in rich SOM soils (Rowley et al., 2017). Beside the limitations of oxygen availability due to water-saturation, and extreme temperature constraints on bacterial and enzymatic activity (Rowley et al., 2017), the most significant preservation mechanisms in mineral soils are SOM aggregation and mineral sorption (Lützow et al., 2006; Kleber et al., 2015). SOM aggregation is a physical process by which SOM is separated from decomposers by occluded organic and inorganic particles aggregate with cementing agents (Six et al. 2002). Aggregation is known to be
promoted both by microbial activity (Chenu and Cosentino, 2011) and inorganic components in the soil matrix (Vitro et al., 2013). The sorption mechanism involves a wide range of reactions between organic compounds and soil minerals. Various clay minerals, oxides, and cations exhibit a high affinity for bonding with organic matter, leading to stabilization (Kleber et al., 2015; Rowley et al., 2017). Recent studies indicate that microbial residue, known as necromass, and depolymerized organic compounds produced by microbes are highly correlated with
mineral-bonded OM, more than complex recalcitrant plant-derived material (Liang et al., 2017; Manzoni & Francesca, 2024).

Pristine peatlands lack these mechanisms (i.e., aggregation and sorption) due to their low mineral content (Leifeld et al., 2012; Mirabito & Chambers. 2023). However, these preservation mechanisms may become relevant after prolonged mineralization of peat soils. Limited knowledge exists about SOM pools and stabilization mechanism
in degraded peat soils. Thus, making it challenging to accurately characterize long-term peat degradation and select suitable models for describing peat SOM loss over time.

To the best of our knowledge, no long-term data or models on peat decomposition are available in the literature, although many studies have been conducted on mineral soils. For instance, multiple long-term Bare Fallow experiments, spanning several decades, have been conducted on mineral soils in cold temperate climate zones
(Barre et al., 2010). These experiments are especially valuable for identifying stable SOM, as they exclude organic matter inputs to the soil - a condition resembling drained peat.

Here we study a peatland site located in the Hula Valley, Israel, which has several important characteristics: (1) The site is located in a warm, subtropical climate; (2) The peat drainage occurred nearly 70 years ago, initiating a peat degradation period; (3) A rewetting of the bottom part of the degraded peat column, occurred three decades



ago, and left the upper peat column drained until present time; and (4) Numerous historical studies were conducted in this site, enabling long-term (66 years) tracking of SOM stock. This research hat two main objectives: First, to evaluate the long-term effect of rewetting on SOM preservation after nearly four decades of drainage in a warm climate. Second, to examine the decomposition rate of nearly 70 years long-term degradation of drained peat and to explore the fate of peat SOM after prolonged mineralization. To achieve this, we integrate

Rock-Eval analysis, short-term respiration experiments, historical geochemical data, and a SOM decomposition model to investigate the long-term effects of drainage and rewetting on SOM degradation and preservation in the Hula Valley peatland.

## 2    Study site

The Hula Valley, Israel (33.1°N, 35.6°E), spans 175 km², of which 28 km² consists of drained marshland that

accumulated peat, now used mostly as cropland. The basin sustained lacustrine environments for the past 4 million years (Bein, 1986), with the last Hula Marsh prevailing from 2.5 to 4 thousand years ago until its drainage in 1957 (Cowgill, 1969; Bein, 1986). The dominant species found in the accumulated peat soil of the historic marsh was Cyperus Papyrus (Bein and Horowitz, 1986). The peat has accumulated in the top section of the sediment, reaching up to 12 meters of peat at its central area (Bein, 1967). This central area is also known as the

'deep peat' area which spans 12 km² (Marish, 1986). In 1957, the Hula Marsh and Lake were drained, primarily for agricultural land use and of reduction of water loss through evaporation. Consequently, the habitat was destroyed, and the exposed peat underwent aerobic conditions, accelerating SOM oxidation in the following years. This led to spontaneous peat fires, erosion, soil loss, subsidence, and nutrient leaching into downstream waterways (Hambright and Zohary 1998; Litaor et al., 2011). In 1994, a major agricultural peat land preservation

project was initiated, in which partial rewetting was implemented to halt soil degradation and fires by means of raising and stabilizing the WTL and frequent irrigation of crops (Hambright and Zohary, 1998). Since then, the water table is managed at approximately -100 cm (Tsipris et al., 2021). Hence, the Hula's deep peat area can be divided vertically into 3 sections: (1) The deep pristine peat section (that was not drained) (2) mid-section of peat drained for 37 years (1957-1994) and then rewetted for the last 30 years (1994-present) and (3) the top peat

which was drained over the last 66 years (1957- present). The ambient air annual mean temperature is ~20°C (Marish, 1986; online data MIGAL, 2023), which classifies the area as a tropical soil respiration zone (Bond-Lamberty et al., 2024). Soil average daily temperature measured at -20 cm depth was ~30°C in summer months (Jun-Oct 2023) and ~15°C in winter months (Dec-Mar 2023), with annual mean of ~22°C (online data MIGAL,



2023). The annual mean precipitation was 610 mm (1987-97; Tsipris and Meron, 1998), and the peat area is used

for cropland which is regularly irrigated during summer months preventing the upper peat from drying. WTL are routinely measured in 19 observation wells and 4 operational drainage channels throughout the Hula Valley, as documented in MIGAL's multi-annual reports over the past 20 years (Tsipris et al., 2021). However, assessing WTL in the drainage period between 1957 and 1994, is more challenging due to limited data availability. To address this, we relied on historical studies. Yaari et al. (1971b) provides WTL data from core samples collected

in December 1970, specifying the WTL for each core (Table 1). A report by Marish (1986) presents a spatial WTL map for winter 1985, indicating areas where the WTL is above or below -120 cm, and notes that no significant changes in WTL have occurred since Yaari et al. (1971b) (Table 1). Therefore, we adopted Yaari et al. (1971b) WTL data to represent the period from post-drainage (1957) until the rewetting in 1994. Further, we choose to follow Yaari et al. (1971b) soil profile's locations for our study and soil core samples collection for

comparison. Table 1 presents the complete dataset of WTL variations from 1957 to 2021 in our examined profiles.

| Table 1. water table levels in cm (-) for below ground surface | | | |
|---|---|---|---|
| **Profile** | **Yaari et al. (1971b)** | **Marish (1986)** | **Monitoring** (2001-2021) mean by annual (±STDV) |
| A* | -80 | Above -120 | -101 ± 18* |
| C | -220 | Below -120 | -226 ± 17 |
| D* | -120 | Above -120 | -101 ± 18* |
| E* | -165 | Below -120 | -101 ± 18* |
| F | -200 | Below -120 | -95 ± 10 |

* Based on data collected from observation well number 13. Locations marked in Figure 1.

## 2.1    Sample collection

Soil samples were collected in January-April of 2023 from the 'Deep Peat' area (Fig. 1). Five different locations were chosen based on the sampling locations of Yaari et al. (1971a) designated there as Crossline B. For

coordinates, see the Supplementary Material (1.8). To ensure that the sampling will include the pristine peat, each core was excavated to a depth of approximately -400 cm (much deeper than the WTL preceding the drainage; Table 1). We used a manual gauge auger (Eijkelkamp©) of either 1 or 0.5 m long cores and of 5 and 2 cm diameter, respectively. The soil sampling from the cores was based on two collection schemes: (1) sampling of





discrete 2-5 gr sample, collected at 10 cm intervals, into pre-weighed 50 ml tubes. These samples were mainly
used for RockEval analysis, (2) sampling of homogeneous soil sections, based on lithology and historic WTL, at
30-60 cm intervals to cotton aired bags. These samples were mainly used for respiration incubation experiments.

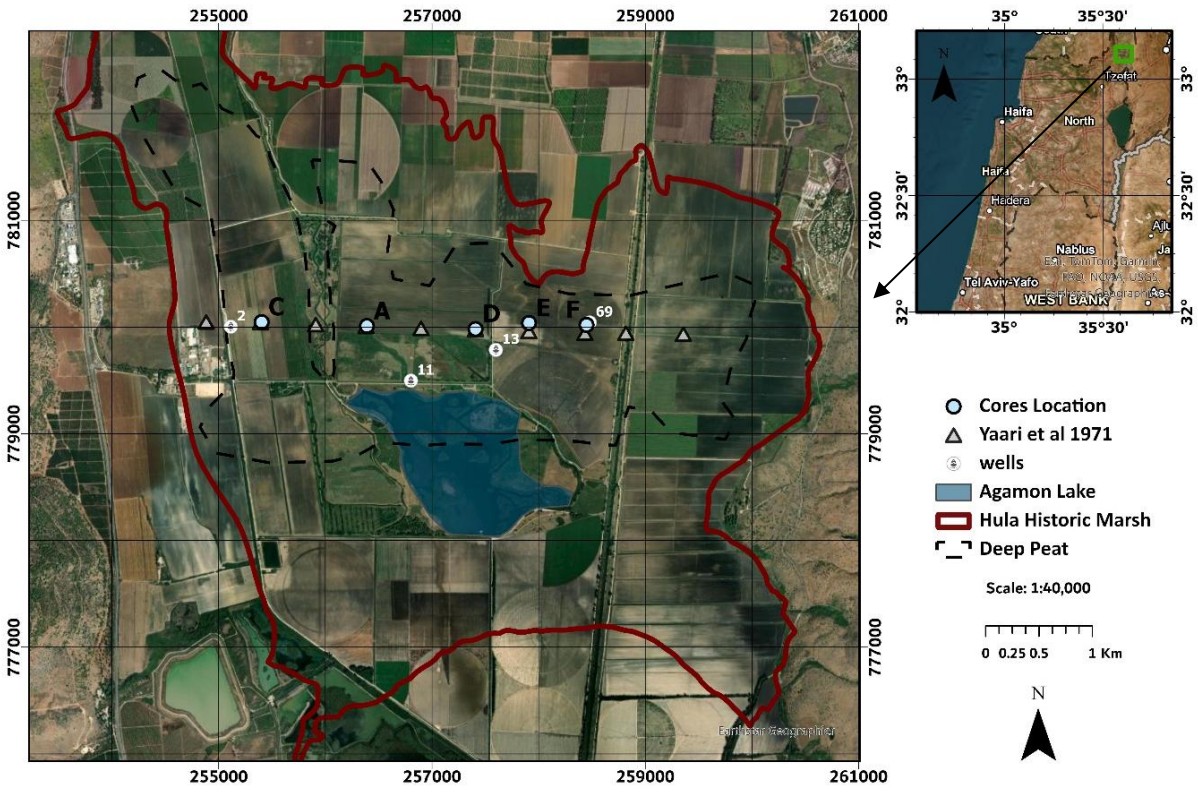

**Figure 1**: Study site satellite map – Hula valley, Israel (coordinates of Israel TM Grid). Polygons of the historic marsh boundaries in
dark red, deep peat area in black dashed line (by Marish 1986), WTL observation wells in white circles, core locations in light blue.
Regional map to the right in WGS 1984 coordinates. This image was made by ArcGIS software and is based on ESRI-provided data.

## 3     Materials and Methods

### 3.1     Rock-Eval Analysis

Analysis of the soil samples for SOM characterization was carried out with the newest Rock-Eval 7S analyzer
(RE-7S, Vinci Technologies). 107 samples of peat from all profiles were freeze-dried, ground, and homogenized
for analysis. About 15-30 mg of ground sample were taken for analysis. The analytical protocol included two



phases: (1) pyrolysis in an inert $N_2$ atmosphere starting at a temperature of 200 °C reaching 650 °C; (2) combustion using air, starting at a temperature of 300°C and rising to 850 or 1200°C when sulfates were also quantified (see below). Both steps heating rate is 25 °C/min, and combustion had a final time of 10 min at maximum temperature before cooling down. The heating cycle is based on Behar et al. (2001) and modified for soils based on Sebag et al. (2016). The gradual heating of the RE-7S promotes a sequential decomposition of organic and inorganic phases according to their thermal stability. Continuous monitoring of the generated products is carried via four different detectors: (1) a Flame Ionization detector (FID) for hydrocarbons (HC), (2) two Infrared (IR) detectors for CO and $CO_2$ and (3) Ultra violate (UV) detector for $SO_2$.

Calibration of C was done using the IFP 160000 standard (Behar et al., 2001). As carbonates are also decomposed during a RE analysis, an empirical separation between organic and inorganic C was previously determined for rocks and mineral soils (Behar et al. 2001). Usually, the organic CO, and $CO_2$ at the pyrolysis phase is determined up to 400°C, and for the oxidation phase, at a local minimum above 650°C. The mineral carbon (MINC) is the remaining fraction of the $CO_2$ and CO curves, calculated between 400-650°C in the pyrolysis phase and approximately between 650-850°C in the oxidation phase. To validate the usefulness of this discrimination, we also applied a loss-on-ignition (LOI) procedure to remove SOM and to identify the correct separation between soil organic carbon (SOC) and mineral carbon (MINC) (see Supplementary Material 1.5). SOC is calculated by summing the hydrocarbon signal (HC), CO, and $CO_2$ curves from both the pyrolysis and oxidation phases. The HC peak, representing carbon in a reduced state, is used to calculate the hydrogen index (HI) of the SOC mg HC gr $SOC^{-1}$. The HC signal was also integrated in 4 parts representing different thermal stability regimes according to Sebag et al. (2016): $A_1$ (200-340°C), $A_2$ (340-400°C), $A_3$ (400-460°C), $A_4$ (460-650°C), which are used to calculate two indices: The R-index (RI) calculated as $(A_3+A_4)/A_{1-4}$ is indicative of the relatively thermal stable fraction of the HC, while the I-index ($\log_{10}[(A_1+A_2)/A_3]$) is indicative for the thermal labile fraction of HC (Sebag et al., 2016). Similarly to the HI, the oxygen index (OI; mg $CO_2$ gr $SOC^{-1}$) represents the $CO_2$ released in the pyrolysis phase (200-400°C), corresponding to oxygen in the OM. The TpkS2 °C is the temperature at the maximum HC signal.

Calibration of sulfur signal in both the pyrolysis and combustion phases was done using pure organic S compound (Di-benzothiophene) and pure pyrite following Cohen Sadon et al. (2022). For real samples that contain multiple S fractions, the organic and pyrite fractions are separated during the pyrolysis phase and can be distinguished by a local minimum at approximately 500°C (Cohen Sadon et al., 2022). However, during the oxidation phase, the residual organic and pyrite S are co-eluted at around 400°C, requiring an empirical relation to deconvolute (Cohen Sadon et al., 2022; Aboussou, 2018). Moreover, some of the pyritic S is retained during



pyrolysis at the presence of carbonates and is released at higher temperature (~700-900°C). Sulfate sulfur
measurement requires high temperatures and is released during combustion above 900-1000°C (Aboussou, 2018).
In this contribution the cutoff between retained S and sulfate signals was set manually to the local minimum
between them, at temperature of ~1000°C.

## 3.2   Soil incubation

Soil samples were oven-dried at 60°C overnight and then stored in the lab. Each profile consists of 6 samples,
each sample was gently grounded, if it contained large aggregates, and sieved to < 2 mm. From each sample, four
splits of 1.5 g portion of soil were weighed to 6 ml vessel, then distilled water was added to 60% of its water-
holding capacity. Water holding capacity was determined for degraded and pristine samples separately. This was
done by saturating 15 gr of soil with 50 ml distilled water and measuring the drained volume of water, using
Whatman 40 filter (8μm). All samples were preincubated for bacterial acclimation for five days at the designated
temperature. The experiments were aerobic and conducted in duplicates under two conditions: (1) for seven days
at 22°C, and (2) for two days at 33°C. During the incubations, gas samples were collected for $CO_2$ and $O_2$
concentration analyses. Samples were taken using glass flasks (~3.7 mL or ~13 mL, depending on incubation
duration) equipped with Louwer™ O-ring high-vacuum valves. Measurements were taken at halfway (t=1) and at
the end (t=2) of the experiment to validate a linear change in gas concentration during incubation. The $O_2$ and
$CO_2$ of the air samples were measured in the laboratory by a closed system (The "Hampadah"; Hilman and
Angert, 2016). The system is based on two analyzers: an infra-red gas analyzer (IRGA) for $CO_2$ measurement (LI
840A LI-COR; Lincoln, NE, USA) and a fuel-cell based analyzer (FC-10; Sable Systems International, Las
Vegas, NV, USA) for measuring $O_2$, and is fully automated. Each sample was incubated and measured in a
known volume, to derive the percentage and concentration of $CO_2$ and $O_2$. Fluxes rate, and reaction constant ($k$)
of $CO_2$ and $O_2$ were calculated for both 22°C and 33°C as shown in Equations 1 and 2, respectively. The apparent
respiratory quotient (ARQ), as shown in Equation 3, represents the ratio of $CO_2$ and $O_2$ fluxes (Angert et al.,
2015). This ratio provides information on the respiratory substrate stoichiometry as well as the underlying biotic
and abiotic processes (Hilman et al., 2022). Temperature sensitivity of organic matter decomposition, expressed
as $Q_{10}$ (Eq. 4), is a metric for assessing the vulnerability of organic matter to temperature increases (Hilasvuori et
al., 2013). $Q_{10}$ was calculated using the rate constant $k$, derived from respiration experiments conducted at 22°C
and 33°C, which represent the annual average soil temperature and its maximal summer temperature in Hula
valley, respectively.



$$(Eq. 1)\ Flux = \frac{CO_{2\ (t=2)}[\mu mol] - CO_{2\ ambient}[\mu mol]}{days * gr\ SOC}$$


$$(Eq. 2)\ k\ constant\ [yr^{-1}] = Flux * 12\ \frac{gr\ C}{mol} * \frac{1gr}{10^6\ \mu gr} * \frac{365\ days}{1\ yr}$$

$$(Eq. 3)\ ARQ = \frac{[CO_{2\ (t=2)}] - [CO_{2\ ambient}]}{[O_{2\ (t=2)}] - [O_{2\ ambient}]} \qquad (Eq. 4)\ Q_{10} = \frac{k(33°C)^{\frac{10}{33-22}}}{k(22°C)}$$

### 3.3 Statistical analysis

Statistical analyses were performed using JMP software (JMP®, JMP Pro 17, SAS Institute Inc., Cary, NC, USA). To determine difference between the peat column sections we used one-way analysis of variance (ANOVA) and Tukey-Kramer test. For unequal variances, we used a Welch's test and nonparametric comparisons using Steel-Dwass test. Significant differences were determined at $P < 0.05$.

The geochemical and respiration data were statistically analyzed based on three peat profile sections: drained, rewetted, and pristine, defined according to their respective historic and current WTL's (Table 1). The drained section included samples from depths of -30 cm to -100 cm, except for profile C, in which it extended to -180 cm. The rewetted section, available only in profiles E and F, ranged from -110 cm to -160 cm in profile E and -110 cm to -190 cm in profile F. The pristine section extended from -200 cm to -400 cm in profiles C, E, and F, while

in profiles A and D, it began at a depth of -150 cm. Samples containing more than 30% carbonate content were excluded from the statistical analysis to avoid incorporating lake mineral horizons, which were primarily found at the bottom of the profile.

### 3.4 Historical Data of SOM

Since its drainage in 1957, the 'deep peat' area of the Hula Valley (marked in Fig. 1) has been studied in numerous research projects, some of which have dealt specifically with SOM loss over time, such as the report by the Marish (1986) and Litaor et al., (2011). To assess the changes in peat soil organic matter (SOM) subjected to 66 years of drainage, we focused on the drained section identified in previous studies with multiple soil profiles (Schalinger et al., 1970; Yaari et al., 1971b; Marish, 1986, Litaor et al., 2003). We constrained our data

compilation to the boundaries of the 'deep peat' area to ensure consistency and avoid significant lithological variations, as well as the impact of soil fires, which mainly occurred south of this region. For depth selections, we





set the upper boundary at -30 cm to minimize the influence of agricultural practices such as soil tillage and root
bias. To focus on the upper, drained peat, the lower boundary was established at -100 cm, which reflects the
prevailing WTL in the region, given the spatial heterogeneity of the WTL. To convert the SOC results to SOM
data, as reported by previous studies, we measured gravimetrically the SOM by removing SOM through
combustion (using $LOI_{400}$ procedure; Nelson and Sommers 1996) to a set of samples. We then correlated the
SOM to SOC and applied a correction factor. Details of this correction are provided in the supplementary
materials (1.5).

### 3.5 Fitted SOM Model

The modelling of organic matter loss through decomposition is a well-studied topic (Manzoni and Porporato,
2009), with various models proposed to represent these processes (Manzoni et al., 2012). Here we chose a model
that comprises of two organic matter compartments (pools), as described by equation 5. These two pools are: 1) a
labile SOM pool decomposing exponentially with reaction constant $k$ and 2) a persistent SOM pool ($SOM_P$),
which is inert, with no decomposition or formation. The $SOM_i$ represents the initial value of the model, P denotes
the fraction of persistent SOM out of $SOM_i$, t denotes time in years, and $k$ serves as the reaction constant of SOM
decomposition. The unknown parameters, $k$ and $SOM_P$, are optimized to minimize the root mean square deviation
(RMSD; Eq. 6), which quantifies the mean difference between model predictions and observed data. Where
$SOM_{data}$ represents field data from various studies, $SOM_{model}$ denotes the simulated values, and n is the number of
observations. The initial SOM ($SOM_i$) value is based on Ravikovitch 1948 work. Historical data and this current
study (five data points, n=5) are used as the observed data ($SOM_{data}$) for RMSD calculation. The model simulates
the long-term decline in organic matter following drainage from 1957 to 2023 (present study).

$$(Eq.5)\ SOM(t)_{[wt.\%]} = \left(SOM_i * (1 - P)\right) * e^{-kt} + (SOM_P)\ ,\qquad P = \frac{SOM_{P\,[wt.\%]}}{SOM_{i\,[wt.\%]}}$$

$$(Eq.6)\ RMSD = \sqrt{\frac{\sum_y^n (SOM_{data} - SOM_{model})^2}{n}}\ , n = 5$$






## 4 Results and Discussion

### 4.1 SOC oxidation and mineralization

Previous research on Hula peat has highlighted a decline in the SOM content within the drained section, attributed to aerobic conditions (Marish, 1986; Litaor et al., 2011). Based on these findings, we hypothesize a

continued reduction in SOM content in the upper drained section and anticipate potential evidence of SOM preservation in the rewetted section, which remained unexplored to date. Additionally, the pristine section, which remained under anaerobic conditions, is expected to exhibit minimal variations in SOM content. Figure 2 shows the measured SOC profiles wt.% from west to east, along with their corresponding WTL as detailed in Table 1. Overall, SOC content in all profiles is lower in the drained and rewetted sections relative to the pristine section.

The profiles can be categorized into two main types: (1) profiles with similar post-drainage and post-rewetting WTLs, resulting in only drained and pristine sections – profiles A, D and C, and (2) profiles with different historical WTLs, creating a rewetted section between the drained and pristine sections – (profiles E and F). In most profiles, there is a clear upward decline in SOC corresponding to the post-drainage WTL (light pink bars, Fig. 2), indicating mineralization and SOC loss due to the lowering of the WTL. For instance, profiles A and C

(type 1) had relatively constant but different WTLs over 66 years of drainage. In both profiles, the depth in which SOC declines above it, aligns with the respective WTLs.

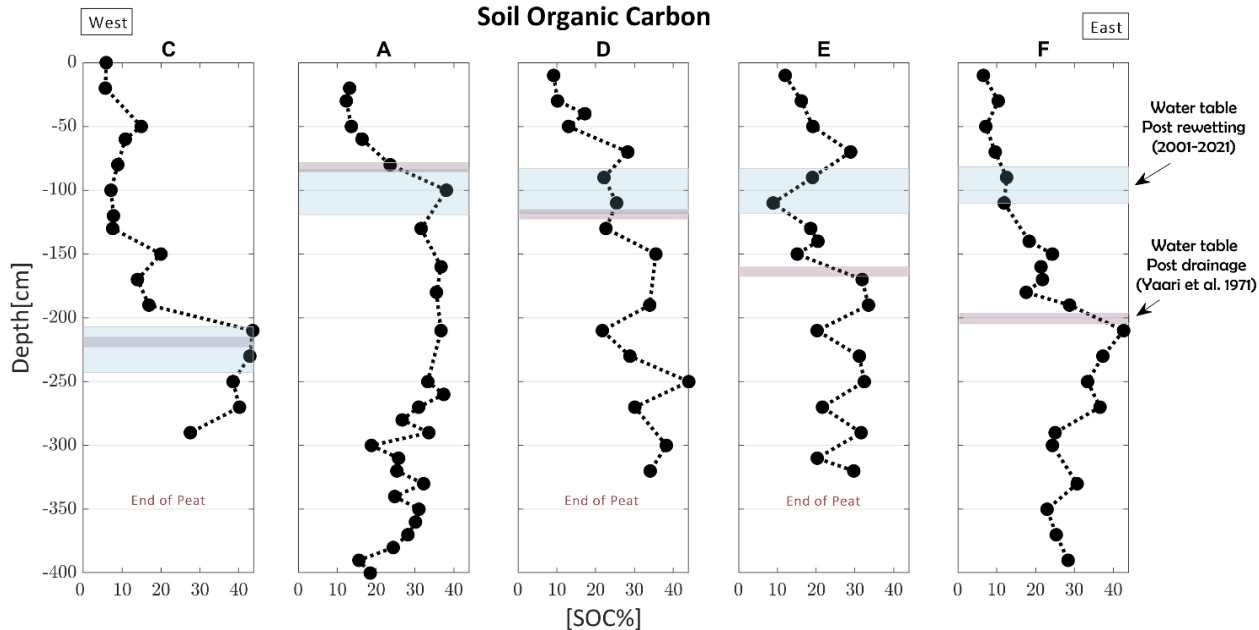

**Figure 2**: SOC profiles (from west to east) of peat cores with respective WTLs - post drainage (purple – after Yaari et al. 1971b) and post rewetting (light blue – MIGAL Monitoring).





The trends in SOM content are further supported by additional organic geochemical data, such as the OI and RI indices. For example, a detailed composite plot of profile F (type 2) identifies three distinct sections (Fig. 3), corresponding to drained, rewetted, and pristine conditions, separated by the historical WTL. Alongside the declining SOC trend, a negative relationship exists between SOC and both OI and RI, indicating that as peat degradation progresses, residual SOC becomes more oxidized, and the proportion of the thermally stable fraction increases. Similar coupling of the historical WTL and the trends of the organic geochemical proxies were found for the rest of the cores, and their composite plots are given in the Supplementary Material (1.4).

Sulfur species, obtained using the RE-7S, were distinguished as detailed in Supplementary Material 1.9. In the drained section, a decline in both organic sulfur and pyrite is evident, while sulfate accumulates, reflecting aerobic conditions (Fig. 3 and Supplementary Material 1.4). In contrast, the pristine and rewetted sections show low sulfate levels but high pyrite concentrations, with a maximum pyrite abundance in the rewetted section. Assuming that pyrite was fully oxidized in the rewetted section during its drainage period (37 years), this peak likely represents a secondary pyrite formation through sulfate reduction under anaerobic conditions following rewetting. The peak in pyrite concentration beneath the recent WTL was observed in most profiles (supplementary material 1.4), suggesting a sulfur recycling mechanism in the drained and rewetted peat sections. Ultimately, pyrite serves as an indicator for both sulfate reduction in the sediment and recent WTL regimes. The corresponding respiration fluxes reveal two key points (Fig. 3 and supplementary material 1.4): (1) respiration fluxes (normalized to SOC) in the drained and pristine sections are similar despite differences in SOC content and chemical properties (as indicated by OI and RI); and (2) respiration fluxes in the rewetted section are twice as high as in the other two sections. These findings are explored further in the next section. pH levels remain mostly neutral but drop to 5.9 in profile F and as low as 3.5 in other profiles. Litaor et al. (2011) also found pH levels of 5-3.8 at a depth of 50 cm in carbonate free horizons, which they interpreted to result from pyrite oxidation, and nitrification.





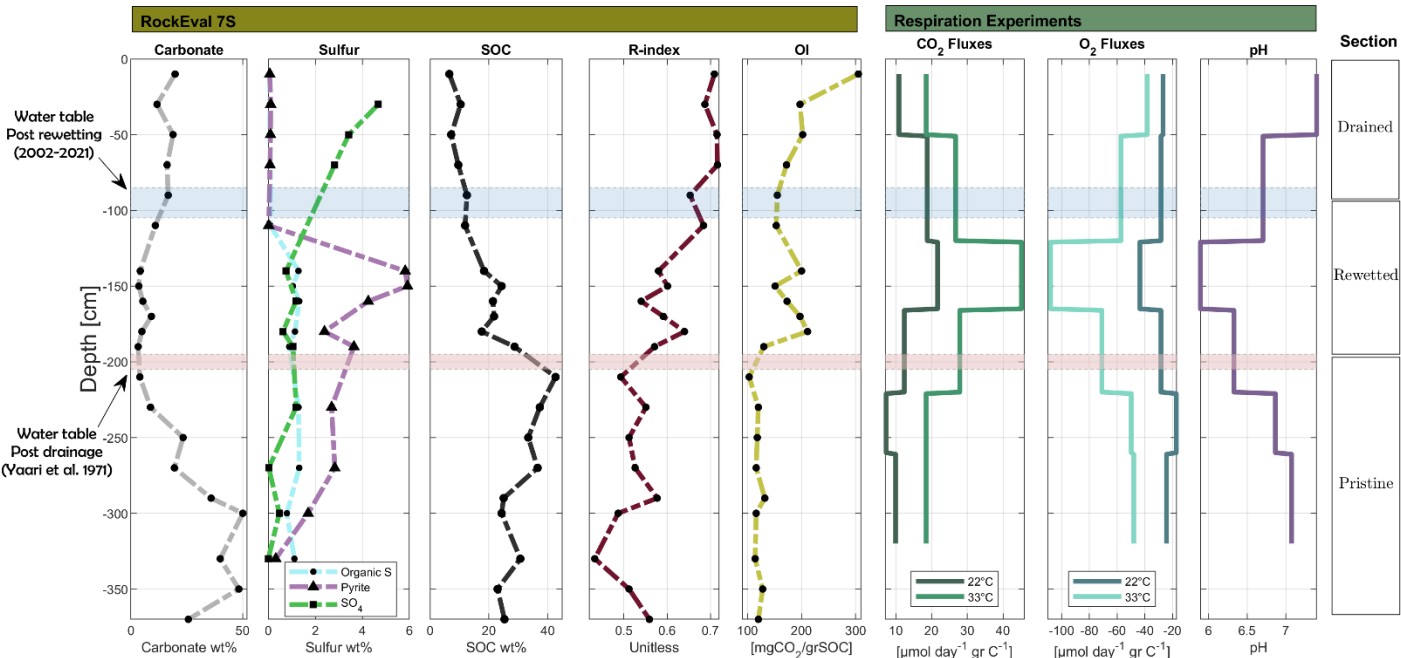

**Figure 3**: Detailed composite plot for profile F, including the historic water table levels.

## 4.2 Average SOC characteristics of drained, rewetted and pristine peat

In the current section we present a statistical compilation of all the soil profiles data. The analyses are made for the predefined three peat column sections: drained, rewetted, and pristine. Figure 4 present the average thermal

analysis indices for each section. Overview of the results and statistical tests for sections comparison are presented in supplementary Tables S1 and S2.

Significant difference ($p < 0.05$) between all three section was found for SOC, TpkS2, RI, I-index and OI (one exception is that the drained OI values is not significantly different from the rewetted). The mean SOC values for the drained, rewetted, and pristine sections are 13 ±1, 20 ±2 and 35 ±1 wt.%, respectively. Thus, it is evident that

66 years of drainage have led to extensive degradation, reducing SOC to approximately one-third of its original pristine value, without considering compaction processes. The rewetted section shows higher SOC content than the drained section, suggesting that degradation ceased or decreased following rewetting. The OI shows a reverse pattern to the SOC, with a gradual decrease in its mean value from the drained to the rewetted and pristine sections (195 ±7, 160 ±11 and 100 ±7 mg $CO_2$ gr SOC$^{-1}$, respectively). This can indicate a partial oxidation of the

SOM due to drainage and a slowdown in oxidation due to rewetting. This observation is further supported by



other SOM thermal indices (Fig.4; correlation in supplementary Fig.S6). Mean TpkS2 values for the drained, rewetted, and pristine sections are 460 ±5, 431 ±7 and 378 ±5 °C, showing the increase of HC peak temperature along with degradation and SOC decline. The mean RI values for the drained, rewetted, and pristine sections are 0.67 ±0.01, 0.59 ±0.02, and 0.49 ±0.01, respectively, while the mean I-index are -0.03 ±0.01, 0.1 ±0.02 and 0.26

±0.01, respectively (Table S1). While the RI and I-index values of the pristine section represent organic soil, those of the rewetted and drained sections fall into the range typical for organic-mineral and mineral soil horizons (Sebag et al., 2016). The RI values of the drained and pristine peatlands further align with the findings by Brown et al. (2023), who reported higher RI values for disturbed peat sites compared to pristine peat sites, all measured from the top 50 cm peat layer (duration since drainage was not specified).

Together, these results demonstrate that with decomposition the residual SOM becomes more oxidized and contains a lower proportion of thermally labile compounds. The increase in thermal stability could be due to a relative increase of litter recalcitrant material (Könönen et al., 2016; Barré et al., 2016), that the residual SOM is minerals associated (Saenger et al., 2015), or both. Moreover, the rewetted section exhibits a higher degree of SOM preservation compared to the drained section, highlighting the impact of rewetting on reducing degradation.

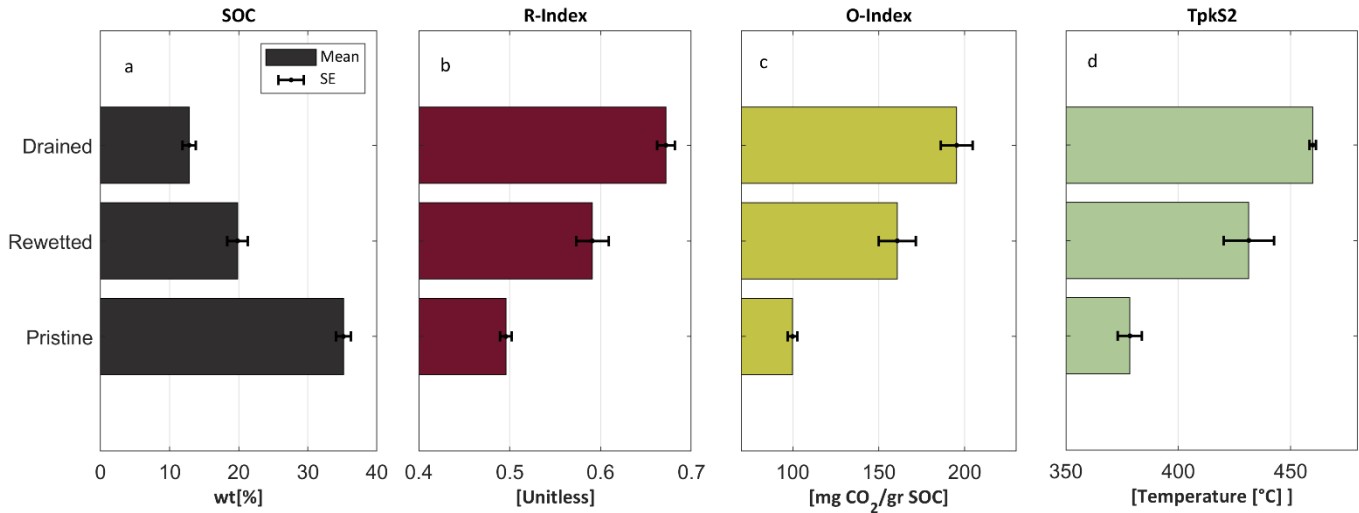

**Figure 4**: Rock Eval analysis results - mean value and standard error of all the profiles from three sections[cm]: drained [30-100] (n=25), rewetted [110-170] (n=10) and pristine [200-400] (n=26). (a) SOC content, (b) R-index, (c) Oxygen index and (d) HC peak temperature - TpkS2.




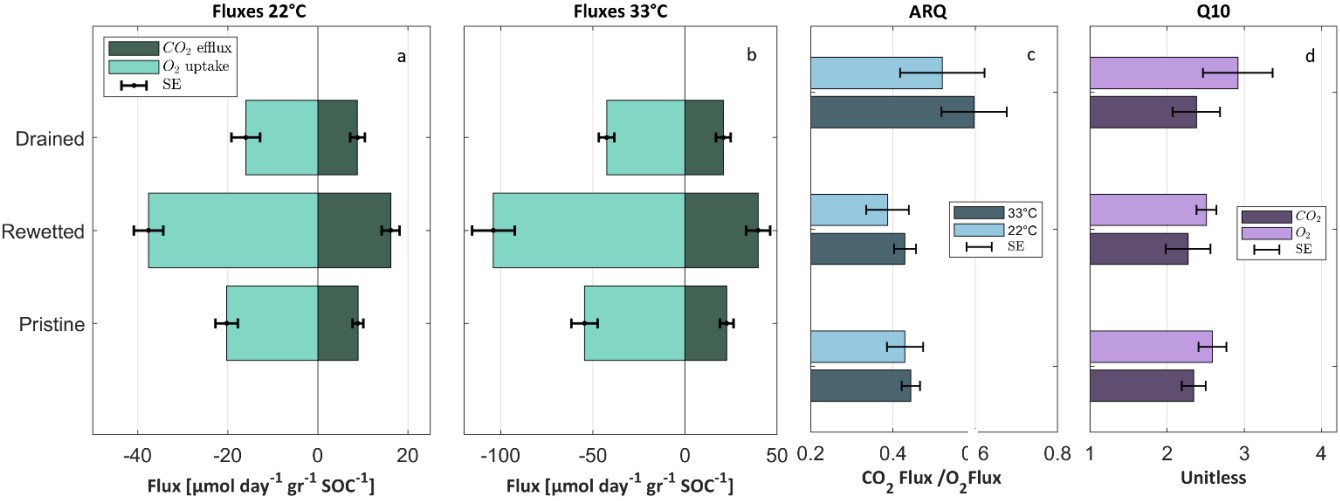

**Figure 5**: Respiration experiment results - mean value and standard error of all the profiles from three sections[cm]: drained [30-100] (n=8), rewetted [110-170] (n=4) and pristine [200-400] (n=12). (a) $CO_2$ and $O_2$ fluxes at 22°C experiments. (b) $CO_2$ and $O_2$ fluxes at 33°C experiments. (c) ARQ -$CO_2$ fluxes divided by $O_2$ fluxes derived from 22°C and 33°C experiments. (d) Q10- derived from $CO_2$ and $O_2$ k constant.

The experimental respiration results were also statistically analyzed in each of the defined sections for all the profiles collectively. Figure 5 displays the mean $CO_2$ and $O_2$ fluxes per gram of SOC at 22°C and 33°C, and their
derivatives indices – ARQ and $Q_{10}$. Mean fluxes of $CO_2$ of the drained, rewetted, and pristine sections at 22°C were 9 ±2,16 ±2 and 9 ±1 μmol $CO_2$ gr $SOC^{-1}$ $day^{-1}$, respectively (Fig. 5a), showing a similar flux for the drained and pristine sections, and approximately a double flux in the rewetted section. This pattern is also evident for the $O_2$ fluxes at 22°C (-16 ±3, -37 ±4 and -20 ±2 μmol $O_2$ gr $SOC^{-1}$ $day^{-1}$) for the drained, rewetted and pristine sections, respectively; Fig. 5a). The same pattern is evident in the 33°C experiments in both $CO_2$ and $O_2$ (Fig.
5b), reinforcing the reliability of this finding. We found similar fluxes in the drained and pristine sections despite the distinctly their different SOC, RI and OI values (Fig. 4). A similar observation was made by Bader et al. (2017), which also found that the degraded peat horizon in cold temperate sites exhibited similar flux per SOC as deeper, less disturbed peat. Moreover, in our study, the rewetted section which has intermediate SOC, RI and OI values (compared with the pristine and drained sections), had the highest respiration fluxes. These results indicate
that SOC content and RE-7S proxies (RI, OI, and I-index) do not directly reflect SOM bioavailability for short-term bacterial respiration.





The first order reaction constant $k$ (Eq. 2) was calculated for aerobic SOC decomposition at 22°C, resulting in -0.037, -0.087 and -0.033 $yr^{-1}$, for the drained, rewetted and pristine sections, respectively. Temperature sensitivity, as indicated by the mean $Q_{10}$ values, for $O_2$ fluxes was 2.9 ±0.4 for the drained section and 2.5 ± 0.4

for the pristine and rewetted sections, with no significant difference between the sections. Mean $Q_{10}$ values from $CO_2$ of across all sections were relativity similar (2.3 ±0.3), with no statistically significant difference (Table S2). Bader et al. (2017) found that the mean $Q_{10}$ (derived from $CO_2$ flux) for drained peatlands in cold temperate sites was 2.5 ±0.1, which resembles our findings despite the different climate.

The ARQ results also show no significant difference between the sections, with mean values at 22°C of 0.6 ±0.05

,0.43 ±0.07 and 0.45 ±0.04, for the drained, rewetted and pristine sections, respectively. In aerobic conditions the expected ARQ value can be indicative of the oxidation state of the decomposed organic carbon compounds (Hilman et al., 2022). Values of ARQ of 1.0 are attribute to sugars which are more labile, while lower values are attribute to compounds such as lipids (0.73) or proteins (0.77), which can derive from necromass and more recalcitrant material (Masiello et al., 2008; Hicks Pries et al., 2020). However, the pristine and rewetted ARQ

results show lower values that are not attributed to any compound and thus may indicate oxygen consumption by partial oxidation of SOM or oxidation of reduced chemical species such as $Fe^{2+}$ or sulfides (Markel., et al 1998).

### 4.3 Long-term SOM decomposition in the Hula's drained peat

The Hula peat exhibits lithological variations both spatially and with depth, resulting from the natural habitat's

dependence on varying water regimes and plant diversity. Additionally, as mentioned above, a decline in SOM of Hula's peat since its drainage was reported (Marish, 1986; Litaor et al., 2011). Here, we evaluate long-term (7 decades) decomposition of peat in the field, by compiling literature data focusing on the top drained section. To this end, studies that included comprehensive data of spatial distribution and sampling depths together with evaluations of SOM of the drained section (30-100 cm) were compiled and analyzed. The SOC data from this

study were converted to SOM using correlations factors for the Hula peat soils determined here as 1.6 for drained samples and 1.8 for pristine samples (Supplementary Fig. S3). The full data set is summarized in Table 2 and illustrated in Fig. 6, alongside our SOM analysis.

During the 66 years of drainage the mean SOM content of the top layer significantly decreased from 68±4% to 21±2% (Fig. 6). This decline was continuous, following an exponential decay pattern, with the most substantial

loss of organic matter occurring during the first three decades, after which the rate of SOM decomposition slowed. Notably, the initial SOM value reported by Ravikovitch in 1948 aligns closely with the SOM of the



pristine section in our study (2023), supporting the concept of preservation under saturated conditions (Fig. 6). Additionally, the SOM value in the rewetted section also conforms to this timeline, suggesting that it has remained mostly unchanged since 1994. These agreements are illustrated by the grey arrows in Fig. 6.

Table 2. Changes in soil organic matter content in the Hula deep peat area since the drainage.

| Year | Study | Mean SOM% ± SE | n | Spatial n | Depth [cm] | LOI Method |
|------|-------|----------------|---|-----------|------------|------------|
| 1948[a] | Ravikovitch 1948 | 68 ± 4 | 4 | 3 | 0-100 | Not reported |
| 1970 | Schalinger et al., 1970 | 48 ± 2 | 30 | 10 | 25-100 | Not reported |
| 1971 | Yaari et al., 1971 | 47 ± 3 | 10 | 9 | 40-110 | Method 1 |
| 1986 | Marish, 1986 | 33 ± 2 | 9 | 4 | 40-100 | Not reported |
| 2000 | Litaor et al., 2003 | 31 ± 4 | 27 | 27 | 50 | Method 2 |
| 2023 | This Study | 21 ± 2 | 20 | 5 | 30-100 | Method 3 |
| 2023 | This Study | 33± 2 | 8 | 2 | 110-170 | Method 3 |
| 2023 | This Study | 64 ± 2 | 23 | 5 | 200-400 | Method 3 |

**Method 1:** LOI at 850°C and carbonate correction following acid digestion. **Method 2:** Acid digestion and LOI at 850°C.

**Method 3**: This study - SOC analysis with RE-7S and LOI at 400°C conversion (Supplementary 1.5).

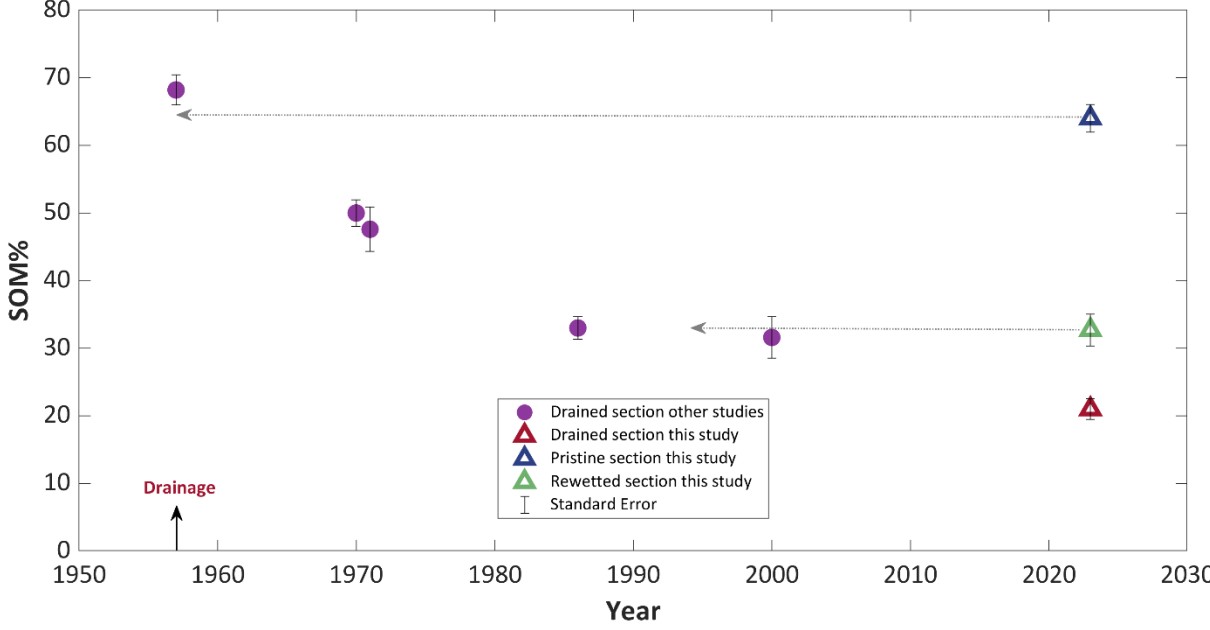

**Figure 6**: Mean soil organic matter (SOM) content of Hula peat since the drainage, based on data from this study (triangles) and historical studies (Purple circles). All from depths of 30-100 cm.




## 4.4    The two-pool SOM model

The decay pattern in Fig. 6 shows an exponential decline, a behavior similar to long-term Bare Fallow experiments of mineral soils (Barre et al., 2010). To better understand the long-term decomposition process of Hula's peat, we explored the observed SOM data with a model. A simple one-box model proved inadequate for
capturing the decomposition process of SOM in mineral soils (Barré et al., 2010; Manzoni et al., 2012; Clivot et al., 2019). The long-term data of Manzoni et al. (2012), based on the Rothamsted Bare Fallow experiment, serves as a good analog to Hula's peat SOM for two reasons: (1) it involves decomposition without significant additional organic matter inputs, and (2) it follows an exponential decomposition pattern over time. Manzoni et al. (2012) suggest that a two-pool model effectively captures long-term soil respiration dynamics, making further
complexity unnecessary. Comparable models were used by Barré et al. (2010) and Cécillon et al. (2021), in which one pool undergoes exponential decay while the other remains stable. Other models account for plant input flux (e.g., the AMG model; Clivot et al., 2019) and leaching of dissolved organic carbon as a pathway for SOM decomposition (e.g., Page et al., 2016, Rixen et al., 2016). Here we assume that plant input to the deep peat (i.e., below 30 cm) is negligible and focus on total SOM loss rather than specific pathways (i.e., respiration vs. DOC
leaching). In addition, although seasonal soil moisture and temperature variations influence SOM decomposition rates, their effects were not addressed explicitly in this analysis. Ultimately, we adopted a two-pool model, where a persistent pool of SOM is initially present and assumed to have a negligible decay rate, while a labile-intermediate pool is consumed with a single reaction constant (Eq. 5).

## 4.5    The SOM Model results

As discussed above the drained and pristine sections of Hula peat have similar respiration fluxes per SOC unit, both for the $CO_2$ efflux and the $O_2$ influx (Fig.5). Similarly, Bader et al. (2017) also found, in their study of peat in temperate-cold climate, that respiration rates per SOC unit in drained and undisturbed peat are comparable. Thus, we suggest that both the pristine and drained peat sections can be characterized by the same reaction constant (see Eq. 5), despite their different organic characteristics. Practically, this similarity allows for a fixed
reaction constant ($k$) to be applied to the labile SOM pool throughout the entire model duration. Also, the similarity of the present-day pristine SOM value to those reported by Ravikovitch 1948 (i.e., pre-drainage) allows us to assume a value of $68 \pm 4$ wt.% for the initial $SOM_i$.





Hence, our two-pool model includes two unknown variables: $SOM_P$ and $k$. With these two degrees of freedom, the model yields a range of possible solutions. The model was run with multiple $SOM_P$ and k values, and the

RMSD was calculated for each run (Eq. 6). Plotting $SOM_P$ versus $k$ and RMSD for all the runs (Fig. S1) reveals that the two unknown variables are related and identifies a single local minimum for the solutions RMSD. In Fig. 7, we illustrate the model results as an envelope of those possible solutions (grey area). This envelope represents model solutions with the lowest RMSD values, defined by a maximum RMSD of 2.2 wt.% of SOM, which ensures that all observed data fit within their standard error. The resulting model envelope ranges for $k$ and $SOM_P$

are -0.03 to -0.042 $yr^{-1}$ and 13–21 wt.%, respectively. To further constrain our model results, we incorporated the reaction constant ($k$) measured by the $CO_2$ fluxes in the short-term respiration experiments at 22°C (which is the mean annual temperature) to the model (Fig. 5a). This reduces the model solution to a single degree of freedom, enabling a direct comparison with the two degrees of freedom model solution. Since the experimental $k$ constant is calculated for the entire SOC content, it must be corrected if $SOM_p>0$ according to the labile SOM $k$ corrected

$=k$ experimental /(1-P)]. Assuming $SOM_P$ of 17 wt.% (the median result of the two degrees of freedom model run), the experimental $k_{corrected}$ constant is -0.044 ± 0.005 $yr^{-1}$ (Fig. 7, green shade). Additionally, we run the model assuming no $SOM_p$ ($SOM_p=0$) and as a result the experimental respiration $k$ constant had no correction factor (k=-0.033 ± 0.004 $yr^{-1}$; Fig. 7, blue line).

As Fig. 7 shows, using the experiment $k$ as an input, and assuming that there is no stable fraction of SOM

($SOM_p=0$), results in a predicted decomposition (blue line) which is much faster compared with the field data. This model run predicts a SOM content in 2023 of approximately 8%, which is inconsistent with the observed value of 21%. In contrast, the model run which used the experimental $k$ and a $SOM_p$ value of 17% (green envelope), effectively explains the field data and aligns closely with the RMSD envelope of the two degrees of freedom run (grey envelope). Therefore, the short-term respiration experiments can be used to bridge the gap

with long-term (70-years) process of peat degradation, at least at this site. In summary, we suggest that peat degradation is a process lasting several decades under a warm climate, during which a fraction of persistent organic matter remains in the soil, even after long-term degradation given the prevailing land-use practices.





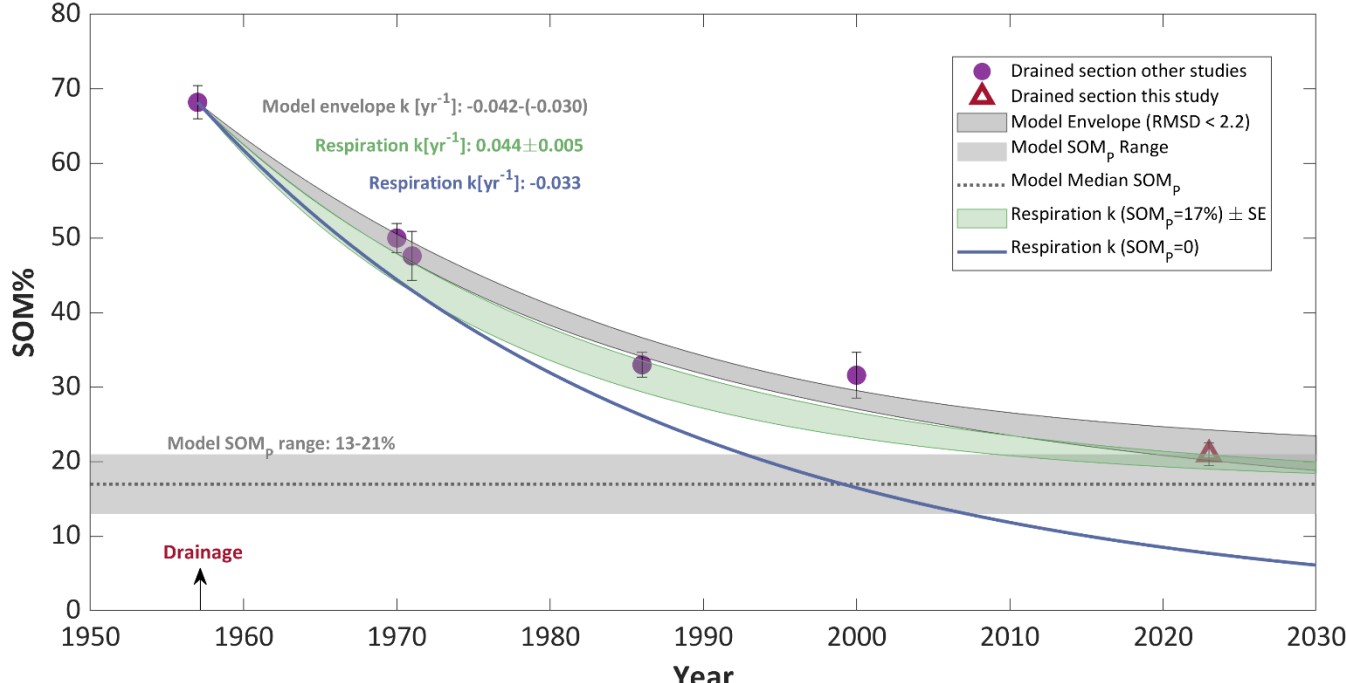

**Figure 7**: Mean soil organic matter content of the upper 1 meter of Hula peat since drainage in 1957, represented by markers. The fitted model result is shown by the grey envelope of possible solutions with RMSD lower than 2.2% SOM. The horizontal grey area and dashed line represent the persistent SOM pool range and median, respectively, predicted by the model. The experimental respiration decay result is represented by the green area (width is indicating the standard error), assuming 17% $SOM_P$. The blue line shows the result of the experimental k with zero $SOM_P$.

## 4.6 The model limitations

It is important to note some limitations: (1) our analysis assumes the drained section has not been significantly affected by the partial rewetting initiated in 1994. While this section technically remains drained after the rewetting project, the increase in water content could influence soil respiration and potentially alter decomposition rates, either enhancing or reducing them. (2) The respiration experiment might overestimate the actual rate since it is measured in optimal water holding capacity (60%), which does not typically occur in natural

settings. We suggest that this might be the reason for the experimental $k$ plot to project somewhat more intense decomposition compared to the observed data (Fig. 7, green shade). (3) Another potential limitation is the possibility of enhanced decomposition in the drained section due to the priming effect caused by root exudates from crops (Fontaine et al., 2004). However, Linkosalmi et al. (2012) found no evidence of a priming effect in





their study of drained peatlands, suggesting that the introduction of labile carbon substrates, such as root
exudates, did not significantly accelerate the decomposition of older, more persistent organic matter. (4) It is
acknowledged that additional SOM inputs and outputs that are not accounted for in the model may contribute to
the overall trend. These may be root carbon contributions or DOC leaching for example. While the model, which
is based on observed data, effectively captures the overall SOM losses and gains, this remains a limitation when
directly comparing it to respiration fluxes. Nevertheless, as pointed out above, we find the model in its current
form - both satisfactory and informative.

### 4.7    Persistent SOM in the Hula peat

While our model analysis suggests the presence of original persistent SOM and further confirms it by including
the short-term experiments' decomposition rate constant, direct evidence for persistent SOM is still lacking.
Recent studies have attempted to quantify persistent SOC using Rock-Eval analysis (Saenger et al., 2015;
Cécillon et al., 2021). Here, we discuss this knowledge regarding SOC persistence in relation to our findings.
Such persistent SOM could have been an original fraction of the peat or formed during the oxidation process or
resulted from a combination of both. Furthermore, the mechanism behind its persistence (e.g., mineral association
or other processes) in drained peat remains unknown.

In addition to our SOM model, the correlation between mean SOC and RI values across sections highlights the
increase in thermal stability of the drained section compared to the pristine section. Figure 8a presents pyrograms
of the Rock-Eval HC signal generated during the pyrolysis ramping. It shows that pristine samples (blue curves)
exhibit two dominant HC peaks, corresponding to thermally labile and stable fractions (Sebag et al., 2016). In
drained samples, these peaks persist but are diminished, with the high-temperature peak less reduced. This
suggests that during drainage, as bacterial activity accelerates organic matter decomposition, the thermally labile
fraction (up to 400°C) is preferentially consumed, leaving behind residual thermally stable SOC, contributing to
higher RI values in drained samples.

The mechanism and evaluation of persistent SOC has been widely studied and debated (Cécillon et al., 2021).
Barré et al. (2016), Gregorich et al. (2015), and Saenger et al. (2015) demonstrate that while persistent SOM has
diverse chemical compositions, it generally exhibits low energy content and hydrogen depletion, making it more
refractory at high temperatures. This aligns with our findings (Fig. 8). Saenger et al. (2015) proposed a thermal
fractionation of SOC using RE analysis, identifying three SOC pools based on the pyrolysis HC curve: labile,
particulate, and mineral-associated SOC - a classification widely accepted in soil science.





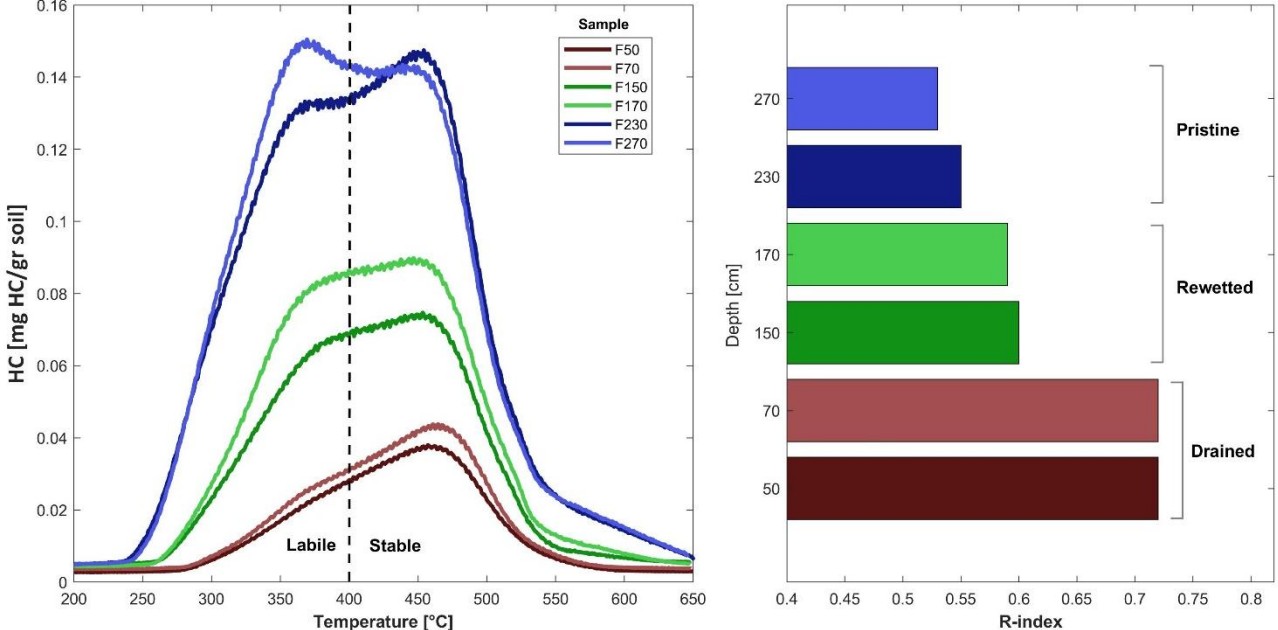

**Figure 8**: (a) FID signal curves of hydrocarbons for pristine, rewetted, and drained samples from profile F, with colour scheme by section type and depth. The dashed line separates labile and stable hydrocarbons, following the method used for RI (Sebag et al., 2016). (b) RI values for the same samples, using the same colour scheme.

Our thermal analysis partially aligns with these findings. The stable SOC HC peak (above 450°C), attributed by Saenger et al. (2015) to mineral-associated SOC, appears in all sections. In pristine samples, it is as high as the labile SOC peak. However, mineral association with SOC is expected to be very low in pristine peat samples (Mirabito and Chambers, 2023). This demonstrates that attributing HC peaks to particulate and mineral-associated SOC presents a challenge in Hula's peat.

A new approach to estimating persistent SOC in soils using Rock-Eval was proposed by Cécillon et al. (2021) through the PARTYsoc model (v2.0). The results indicate that, unlike the thermal indices we used above, the predicted stable SOM by the PARTYsoc model showed only a weak negative correlation with SOC content ($R^2 = 0.25$; supplementary material 1.6). The model estimates of the stable fraction was multiplied by the SOC content, yielding a mean persistent SOC of $13.2 \pm 0.8$ wt.% in the pristine section and $6.1 \pm 0.5$ wt.% in the drained section. However, persistent SOC is not expected to decline so fast during 66 years of peat decomposition since it is defined as resilient for more than a century (Cécillon et al., 2021), suggesting a discrepancy in these results.





Our findings show that the estimation of the PARTYsoc v2.0 model or pool separation by Saenger et al. (2015) is inadequate for the Hula's peat. However, while RI does not deliberate quantification of persistent SOC, it could serve as a practical and effective tool for analyzing and comparing degraded peat SOC, both in research contexts and for peatland management.

**4.8 Estimation of annual CO$_2$ emissions**

Respiration models and global annual emission estimates typically incorporate factors such as peat SOM content, bulk density (BD), and WTL. However, these models often neglect the critical factor of time since drainage, which significantly influences emission rates (Hooijer et al., 2010; Tubiello et al., 2016). To emphasize the impact of drainage duration on carbon fluxes, we compared respiration data from pristine and drained sections of

the Hula, representing the onset of drainage and nearly 70 years of degradation, respectively. The average annual carbon emissions flux from the pristine and drained peat sections of the Hula at 22°C, derived from the short-terms experiments (Flux per gr soil; Eq.1), are $12 \pm 1$ and $4.7 \pm 0.9$ mg C gr soil$^{-1}$ yr$^{-1}$, respectively. Additionally, the average annual carbon loss rates derived from the observation-based long-term model (Fig. 7, grey area), calculated as (SOC$_{intial}$ - SOC $_{(10yr)}$)/10, are $9 \pm 1$ and $1.4 \pm 0.2$ mg C gr soil$^{-1}$ yr$^{-1}$ for the 1$^{st}$ and current 7$^{th}$

decades, respectively. These two independent methods demonstrate the drastic difference in CO$_2$ fluxes between early and advanced stages of peat degradation. The comparison between these fluxes is presented in Fig. 9.

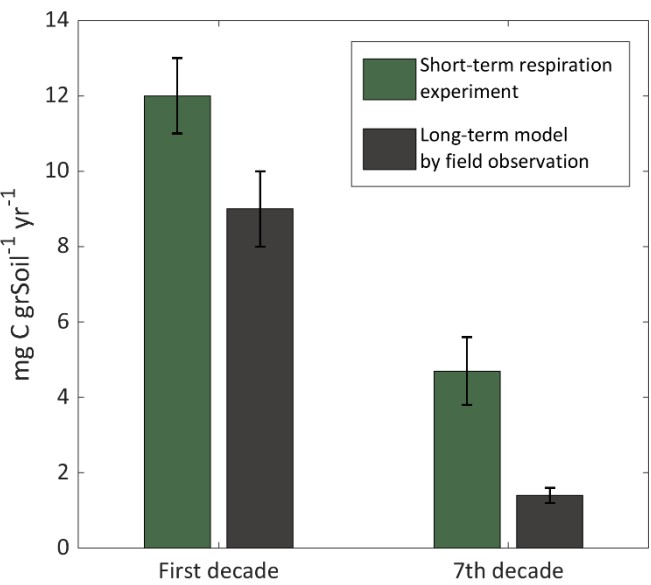

**Figure 9**: Average annual fluxes of CO$_2$ from peat soil, derived from short-term experiment (green) and long-term field model (black).



The IPCC and global models estimate $CO_2$ emissions from drained tropical peatlands at $73 \pm 14$ tons $CO_2$ ha$^{-1}$ yr$^{-1}$ (Hiraishi et al., 2013; Tubiello et al., 2016). To compare our results with flux rates from warm climate regions, we calculated a theoretical $CO_2$ emission value for a 100 cm drained peat column, using our experimental fluxes at 22°C (annual average) and bulk density of 0.22 g cm$^{-3}$ (Dansberg et al., 1973). The estimated $CO_2$ emission is $94 \pm 10$ tons $CO_2$ ha$^{-1}$ yr$^{-1}$, reflecting emissions during the initial decades after drainage, and is comparable with the IPCC estimates. After 66 years of degradation, assuming 70% soil shrinkage (Yaari et al., 1971b) and an increase in BD to 0.55 g cm$^{-3}$ (Orlov-Levin and Meron, 2005), the current estimated $CO_2$ emission from the drained section is $28 \pm 5$ tons $CO_2$ ha$^{-1}$ yr$^{-1}$.

## 5    Conclusions

Our observations show that active land management that includes partial rewetting of once drained peat column significantly mitigates $CO_2$ emissions and organic carbon loss, demonstrating its effectiveness even after prolonged drainage in a warm climate. The respiration results highlight the vulnerability of the rewetted peat horizon to re-exposure to air, showing the potential of enhanced fluxes, twice as high relative to the drained and pristine sections (per SOC). We also found a peak in pyrite concentration beneath the recent WTL, and sulfate accumulation in the drained section, which indicates sulfur recycling and anaerobic conditions. Short-term respiration experiments reveal that the drained and pristine sections share similar first-order reaction rates ($k$) for SOM decomposition, and that these experiments successfully describe the observed long-term SOM decomposition in the warm-climate Hula's peat. Model analysis, supported by those experimental results, indicate that approximately 13–21 wt.% of the ~68 wt.% initial SOM persists as a stable fraction in the drained peat, emphasizing the role of the persistent organic pool in stabilizing carbon stocks. However, the mechanism of persistence SOM formation in degraded peatlands requires further investigation. The use of a thermal analysis (Rock-Eval) to characterize the OM properties, coupled with short-term aerobic respiration experiments is shown here to be useful in bridging the gap to the field scale, to the modelling of long-term decomposition of peatlands, and to estimating their $CO_2$ emissions. The timeline of drained peat SOM decomposition shows that the first few decades are crucial, as SOM declines rapidly, resulting in high $CO_2$ emissions. Future peatland restoration projects, especially in warm climates, can use this site as an analogue to make more informed decisions for the priorities for sites rewetting.



**6    Acknowledgments**

We would like to express our gratitude to the Hula Valley Monitoring Program, operated by the MIGAL Institute, Tel-Hai College, and the Israel Water Survey, for their valuable collaboration and the availability of their data online. We also thank to Michael Litaor and Oren Reichman for providing access to their field data. We would like to acknowledge Tal Weiner, Roee Katzir and Itay Eyal for their assistance in our field work.

**7    Code availability**

The code for the peat SOM Model can be accessed in this DOI link: https://doi.org/10.5281/zenodo.15635973 .

**8    Data availability**

Data of Rock-Eval 7 analysis and respiration experiments can be accessed in this DOI link:
https://doi.org/10.5281/zenodo.15635973 .


**9    Author contribution**

The Investigation was done by GS. AA, RG, and YR conceptualized the research, were in charge on the methodology, and supervised GS. GS wrote the original draft , and AA, RG and Y reviewed and edited it.

**10    Competing interests**

The authors declare that they have no conflict of interest.

**11    Sample availability**

Samples are stored in the Geological survey of Israel and can be accessed if needed, please reach the author for more information.

**12    Disclaimer**

OpenAI was used solely to improve the grammar and phrasing of the text. The authors declare that no other artificial intelligence or machine learning tools were employed in this study.



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
