# Peer review of "Long-term Effects of Drainage and Rewetting on the Degradation and Preservation of Peat Organic Matter in Warm Climate"

_EGUsphere, 2025_

## Community Comment (CC1)

We thank the reviwer for his important comments, that will help to improve the manuscript. Below is the reviwer comments (in black) and our response and the action we will take (in blue).

**General comments:**

This paper presents an interesting work conducted on a peatland in Israel. The place was drained to be used for cultivation, then rewetted when problems (fires, erosion, etc.) began to arise. The authors sampled five 4-meter cores in the deepest part of the peatland, and identified three different parts: drained, rewetted, and pristine peat. They conducted Rock-Eval thermal analyses to characterise the thermal stability, stoichiometry and properties of the peat. The results show that rewetting the peatland clearly helped reduce SOM loss, although the rewetted part does not come back to its anterior, pristine state.

My main concern regarding this work was whether it is robust to use Rock-Eval to analyse peat; we know that Rock-Eval shows some limits with highly organic compounds (e.g. litter), which most probably apply to the case of peat. This problem has been addressed in Supplement: the authors took the precaution to investigate this question with LOI procedure. I would still be very cautious when applying Rock-Eval to such organic soil, however on this specific case the results are clear and consistent. This paper is a nice addition to the peatlands knowledge.

We agree with the reviewer that Rock-Eval can has some limitations in evaluating the SOC content. Specifically, there are two issues that need to be addressed:

- 1. Compared to rocks, the default cutoffs between organic and inorganic C in soils need to be adjusted, otherwise SOC can be underestimated by up to ~10% (relatively), and mineral C, SIC, is overestimated (Stojanova et al., 2024). While we did not apply the correction method of Stojanova et al. (2024), we did re-evaluated the default cutoffs between organic and inorganic C using LOI as the reviewer states. We prefer our approach, as the nature of the samples in the present study differs from those in Stojanova et al. (2024). Additionally, the low SIC in the present study, for most samples, should minimize any discrepancy in inaccurate cutoffs between SOC and SIC.
- 2. Fresh plant tissues can result in under-estimation of SOC, probably because of inaccuracy of the FID detector of the Rock-Eval. However, for diagenetic soils multiple studies have shown that the Rock-Eval is producing meaningful data both for peat and organic-soils with SOC reaching up to 40 wt.% (e.g., Disnar et al, 2008; Marchand et al., 2008; Sebag et al, 2016; Brown et al., 2023). Specifically, the work of Marchand et al. (2008) compared SOC of Rock-Eval to total C using LECO combustion in the absence of carbonates, and the comparison was almost in full agreement up to ~ 20% SOC. While we cannot exclude some underestimation of the Rock-Eval SOC, the correlation between SOC and LOI in the current study (Fig. S3) further support that inaccuracy of the Rock-Eval is minimal.

Hence, we conclude that the use of the Rock-Eval, despite potential limitation, produced valuable and quantifiable data for discussing peat degradation and CO2 emissions.

**References:**

Brown C, Boyd DS, Sjo"gersten S, Vane CH (2023) Detecting tropical peatland degradation: Combining remote sensing and organic geochemistry. PLoS ONE 18(3): e0280187. https://doi.org/10.1371/journal.pone.0280187

- Disnar JR, Jacob J, Morched-Issa M, Lottier N, Arnaud F. Assessment of peat quality by molecular and bulk geochemical analysis: Application to the Holocene record of the Chautagne marsh (Haute Savoie, France). Chemical geology. 2008 Aug 30;254(1-2):101-12.
- Marchand C, Lallier-Vergès E, Disnar JR, Kéravis D. Organic carbon sources and transformations in mangrove sediments: a Rock-Eval pyrolysis approach. Organic Geochemistry. 2008 Apr 1;39(4):408-21.
- Sebag D, Verrecchia EP, Cécillon L, Adatte T, Albrecht R, Aubert M, Bureau F, Cailleau G, Copard Y, Decaens T, Disnar JR. Dynamics of soil organic matter based on new Rock-Eval indices. Geoderma. 2016 Dec 15;284:185-203.
- Stojanova, M., Arbelet, P., Baudin, F., Bouton, N., Caria, G., Pacini, L., Proix, N., Quibel, E., Thin, A. and Barré, P., 2024. A validated correction method to quantify organic and inorganic carbon in soils using Rock-Eval® thermal analysis. *Biogeosciences*, 21(18), pp.4229-4237.

**Specific comments:**

L135: does that mean the WTL was at surface level before drainage? Do we have information on this? (I understand 'old' information is scarce.)

Our understanding of the water table prior to the drainage of the marsh, during the period of the peat accumulation, is derived from a documented expedition by Jones (1940) and hydrological data that were documented by the hydrological survey and published in a book by Dimentman et al. (1992). Jones reported that the characteristic water table in the marsh was 0.5-2 m above surface in the main papyrus stands. He also mentioned that in places, during the summer months (probably in the northern sections), the peat was exposed and water table was found 0.5-1 m below surface. These data align well with monitoring data of the hydrological survey that recorded the water table absolute elevation since 1935 and brought by Dimentman et al. (1992). These data show that during most years the water level fluctuated in a range of 1.0 to 1.5 m, with corresponding total flooded area between ~15 km² in summer (August-September) and ~45 km² during the winter floods (January-March). Relevant data added in the manuscript, L132.

Dimentman Ch., Bromley H.J., and Por F.D., 1992. Lake Hula, Reconstruction of the Fauna and Hydrobiology of a Lost Lake. The Israel Academy of Science and Humanities, Jerusalem, 170 pp. Jones, R. F. (1940). Report of the Percy Sladen Expedition to Lake Huleh: A Contribution to the Study of the Fresh Waters of Palestine: The Plant Ecology of the District. Journal of Ecology, 28(2), 357–376. https://doi.org/10.2307/2256234

L198: an illustration, similar as Fig.1 in Cécillon et al. (2018), could be useful to visualize how you cut/sum your signals.

Done. As, the purpose of this manuscript is not method development, we add this figure in the supplementary material. We added the CO curves to the original Fig. S4 and added vertical dashed lines to indicate the cutoff between organic and mineral carbon. The figure is cross-referenced in the corrected manuscript.

L210: why consider only the CO2 emitted during pyrolysis, and not CO? Same, why disregard oxygen emitted as both during oxidation? More generally, you chose to follow Behar et al. (2001)'s definition of oxygen index, taking CO2-carbon into account, rather than more recent definition focusing on oxygen only (with stoichiometric correction as in Cécillon et al., 2018; Saenger et al., 2013; Delahaie et al., 2023). Did you consider both definitions before choosing Behar's?

We agree with the reviewer and will consider the CO in the total OI of the revised manuscript.

L245: the explanation for why 22°C and 33°C could figure above, L231, so that we immediately understand why you chose these.

We will add these explanation there.

L249: I think the equations should be rewritten formally so as to only contain variables, not mixed up with units. Describe the variables and their units above or below.

Done. The units for the flux and k are written near their definitions above the equations and were removed from the equations. Note that the units in the right side of Eq. 2 were left to explain the calculation.

L261: the explanation as to why start at -30 cm and not above could be there instead of L277.

L263: why isn't there a rewetted section in the cores A, C, and D? Perhaps I missed the explanation, but I don't understand why the rewetting seems to not have 'worked' everywhere.

This is because the water table post drainage and the water table post rewetting were similar in depth (gray and light blue shaded area, Fig. 2) in those cores, limiting the development of "rewetted sections". We clarify this in the revised manuscript (L262): "A rewetted section is absent from the other cores as the WTL's post-drainage and post-rewetting were very similar (Fig. 2, purple and light blue shaded areas)".

L288: usually, when talking about 'persistent', it is good to precise which duration you are referring to, as it cannot be forever: does it persist for decades? Centuries, millennia?

We agree with the reviewer. However, the persistence of the OM cannot be readily evaluated. Rather, we can state that the fraction of persistent OM remains stable for a duration much longer than the degradation period observed and modeled. That is, much longer than 66 years. We clarify this in the revised manuscript (L288): "a persistent SOM pool (SOMP), with decomposition rates that are significantly lower than the observed duration of degradation for the labile pool". And in L471: "Given the degradation duration of the labile SOM was of the order of 70 years, we estimate that the persistent SOM should be stable for several centuries given similar environmental conditions".

L356: there is debate on the significance of TpkS2, as the peak is not always related to the quantity of hydrocarbons evolved during the whole process (you can have a very short peak at the beginning while most of the matter evolves later). Did you have a look at other indices, such as T90\_HC\_PYR, the temperature at which 90% of hydrocarbons have evolved (as described in Cécillon et al. (2018) for instance)?

We agree with the reviewer that this peak is not always correlated with the quantity of HC evolved, especially as the FID is not a symmetric Gaussian peak in soils and peat as in rocks. Specifically in this study, the FID signal of the pristine peat is composed of two overlapping peaks of similar size (Fig. 8). With degradation, the more labile peak diminishes at a higher rate. The shift in TpkS2 resembles this relative change, and hence we gave it as an example how each peat section is geochemically distinguished.

L459: sentence unclear; perhaps the subscript disappeared, it would make more sense with it...

Corrected. '[' was missing. The corrected sentence is: "Since the experimental k constant is calculated for the entire SOC content, it must be corrected if  $SOM_p>0$  according to the labile  $SOM_k$  corrected k experimental k corrected sentence is: "Since the experimental k constant is calculated to the lability k corrected k experimental k corrected sentence is: "Since the experimental k constant is calculated to the lability k corrected k experimental k corrected sentence is: "Since the experimental k constant is calculated to the lability k corrected k experimental k corrected sentence is: "Since the experimental k corrected sentence is: "Since the experimental k corrected is k corrected sentence is: "Since the experimental k corrected sentence is: "Since the experim

L485: also, the priming effect would probably not extend much under the root depth, which you excluded by starting at -30 cm.

We agree with the reviewer. The following sentence was added to the revised manuscript L485: "Furthermore, in the model the top 30 cm were excluded, minimizing this potential effect".

L493: did you consider radiocarbon analyses?

We did not consider radiocarbon analyses, and we are not certain that this will give a definite answer to the source of the persistent OM. As discussed in the original MS, the persistent OM can be generated with the chemical changes of the peat due to its decomposition. Even if it was there originally, as considered by the model, it can differ from the labile OM because of its interactions with minerals, or because of its chemical structure. In other words, the persistent OM can be of similar age as the labile OM.

L526: the PARTYSOC model has some limitations, even in its v2 form. In particular, soils with a high SOM should not be treated with this model, as it has never been trained nor tested on such data. Agree, in fact our result agree with this.

Technical corrections:

General: check for grammar, non-verbal sentences, etc.

General: the abbreviation for gram is g, not gr.

L313: 'purple' as in the caption, not light pink. Maybe homogenise the color with other figures...

All technical correction will be done.

**References:**

Behar, F., Beaumont, V., and Penteado, H.L.: Rock-Eval 6 Technology: Performances and Developments, Oil & Gas Science and Technology – Rev. IFP, Vol. 56 (2001), No. 2, pp. 111-134, https://doi.org/10.2516/ogst:2001013, 2001.

Cécillon, L., Baudin, F., Chenu, C., Houot, S., Jolivet, R., Kätterer, T., Lutfalla, S., Macdonald, A., van Oort, F., Plante, A. F., Savignac, F., Soucémarianadin, L. N., and Barré, P.: A model based on Rock-Eval thermal analysis to quantify the size of the centennially persistent organic carbon pool in temperate soils, Biogeosciences, 15, 2835–2849, https://doi.org/10.5194/bg-15-2835-2018, 2018.

Delahaie, A. A., Barré, P., Baudin, F., Arrouays, D., Bispo, A., Boulonne, L., Chenu, C., Jolivet, C., Martin, M. P., Ratié, C., Saby, N. P. A., Savignac, F., and Cécillon, L.: Elemental stoichiometry and Rock-Eval® thermal stability of organic matter in French topsoils, SOIL, 9, 209–229, <a href="https://doi.org/10.5194/soil-9-209-2023">https://doi.org/10.5194/soil-9-209-2023</a>, 2023.

Saenger, A., Cécillon, L., Sebag, D., & Brun, J. J. (2013). Soil organic carbon quantity, chemistry and thermal stability in a mountainous landscape: A Rock–Eval pyrolysis survey. Organic Geochemistry, 54, 101-114.

---

## Author Comment (AC1)

We deeply thank the reviewer for his important comments, that will help to improve the manuscript. Below is the reviewer comments (in black) and our response and the action we will take (in blue).

**Overall view**

This study presents valuable and somewhat rare long-term information from the Hula Valley peatland in Israel, tracking almost seventy years of drainage followed by three decades of partial rewetting. It provides valid evidence that drainage leads to rapid carbon loss in warm-climate peatlands and that partial rewetting can slow this degradation. The data and methods are solid, but the manuscript would benefit from clearer terminology, a more careful interpretation of thermal indices, and a balanced discussion of uncertainties.

I believe it will interest readers studying soil carbon dynamics, restoration of degraded peat, and greenhouse gas mitigation. Still, some interpretations are too confident for the evidence provided, and a few sections need simpler explanations. In particular, the differences between thermal 'stability', biological 'resistance', and actual field-scale preservation should be made clearer. Some of the experimental assumptions and terminology e.g., 'tropical' and 'warm-temperate' also need amendment/correction.

**Specific Comments**

**1. Climate description**

The site is described several times as 'tropical'. With an annual mean temperature of about  $20~^{\circ}\text{C}$  and  $\sim 600~\text{mm}$  precipitation, the Hula Valley should be better classified as warm-temperate or subtropical. The wording should reflect this to avoid confusion with true tropical peatlands.

We agree, and the climate definitions will be changed accordingly.

**2. Definition of 'pristine' peat**

The deep layer is called pristine, but it has been under the same area for decades, with drainage above it. Downward oxygen diffusion or chemical change is possible. Please clarify whether redox or sulfur data confirm that it truly remained unaffected.

Firstly, the OM content pre-drainage was homogenous throughout the peat column. Further, the OM content we found at the deep part is comparable to that of the peat pre-drainage. In addition, the deep section has minimal sulfate and high reduced sulfur concentrations (see Fig. 3), reinforcing that oxidation of pyrite (and hence of OM), was limited. Hence, we conclude that oxidation of the deep peat is insignificant. This will be further clarified in the revised version.

**3. Rock-Eval results and interpretation**

The Rock-Eval data are strong, but some interpretation goes beyond what this method can tell. Higher RI or TpkS2 values show greater thermal stability, not necessarily biological resistance. The discussion should separate these two ideas.

**We agree with the reviewer and this will be clarified in the revised version.**

The figure suggests some overlap in OI values between sections, so 'significant differences' should be checked.

We agree with the reviewer and this will be clarified in the revised version.

**4. Respiration experiments**

The incubation tests at 60 % water-holding capacity probably exaggerate natural respiration rates. The authors note this, but the limitation should be discussed more directly.

We will explain this choice of WHC value more clearly in the revised version. This is a value commonly used, intended to be associated with the maximum respiratory rate. Of course, at the field, the drained part is often drier, and sometimes saturated.

The result that rewetted peat produced the highest CO2 flux is surprising. Possible reasons could be the temporary oxygen exposure, recent organic inputs, or sulfide oxidation but it should be acknowledged rather than presented as a steady condition.

**Agree. We will discuss this further in the revised version.**

The ARQ values below one likely reflects partial oxidation or iron/sulfur reactions; this is interesting but needs a sentence of explanation and a reference.

This is indeed part of the explanation. We will discuss this further in the revised version.

**5. Two-pool SOM model and Persistent SOM fraction**

The model is appropriate and well fitted, but it assumes no new organic input and no loss from the 'persistent' pool. Because this area is farmed, these assumptions are not fully realistic. Please mention how they might affect the result.

To minimize OM addition from the top surface, we excluded the 30 top cm from the sections. We will discuss the validity of this assumption, as well as the effect of OM addition from the surface in the revised version.

The estimate of 13- 21 % persistent SOM is plausible. However, the statement that other models like PARTYsoc are 'inadequate' sounds dismissive. It would be better to briefly explain why this method might not fit peat with high organic content.

We agree with the reviewer and this will be clarified in the revised version.

Plus, the metadata (codes and Excel files), show reasonable internal design/structure but mostly represent the drained condition; the distinct 'rewetted' and 'pristine' patterns claimed in the manuscript are not strongly evident here. Variability within the drained layer and method heterogeneity in historical SOM data could explain much of the reported differences.

**We don't understand this comment, please clarify.**

**6. Sulfur and pyrite**

The finding of pyrite peaks below the water table is a good indicator of anaerobic conditions. This part is important and could be discussed in simpler terms: it shows that rewetting successfully re-established reducing conditions.

We agree with the reviewer and this will be clarified in the revised version.

**7. CO2 emission comparison**

The conversion from lab fluxes to tons CO2 ha-1 yr-1 involves bulk-density and shrinkage assumptions that vary widely. A short uncertainty range would make the comparison to IPCC factors more convincing.

We agree with the reviewer and this will be clarified in the revised version.

**Other suggestions**

**Figures and tables**

Figure 1 should mention land use or rewetting zones.

Color schemes for drained, rewetted, and pristine sections should be consistent across all figures.

Table 2 could list the exact loss-on-ignition temperature used in each historical dataset.

**The figures will be modified in the revised version.**

**Typo/grammar notes**

Typo line 120: 'hat two main objectives' should be 'had two main objectives.

Ensure consistent units (wt %, °C, etc.).

Verify all references, especially Manzoni & Francesca (2024) because this looks incomplete.

Some very long sentences in the discussion section should be shortened for easier reading.

Thanks and Good luck.

Citation: https://doi.org/10.5194/egusphere-2025-2763-RC2